# Effect of an illegal open dump in an urban forest on landscape appreciation

**Ernest Bielinis**[1]*, **Emilia Janeczko**[2], **Natalia Korcz**[3], **Krzysztof Janeczko**[4], **Lidia Bielinis**[5]

**1** Faculty of Agriculture and Forestry, Department of Forestry and Forest Ecology, University of Warmia and Mazury in Olsztyn, Olsztyn, Poland, **2** Department of Forest Utilization, Institute of Forest Sciences, University of Life Sciences in Warsaw, Warsaw, Poland, **3** Department of Natural Foundations of Forestry, Institute of Soil Science and Environment Management, University of Life Sciences in Lublin, Lublin, Poland, **4** Department of Forest Management Planning, Dendrometry and Forest Economics, Institute of Forest Sciences, Warsaw University of Life Sciences, Warsaw, Poland, **5** Faculty of Social Sciences, Department of Social Pedagogy and Methodology of Educational Research, University of Warmia and Mazury in Olsztyn, Olsztyn, Poland

* ernest.bielinis@uwm.edu.pl

**Data Availability Statement:** The data underlying the results presented in the study are available from the Harvard Dataverse repository (https://doi.org/10.7910/DVN/53VSL2).

## Abstract

Rubbish in a forest environment is a great threat to this ecosystem, but this threat may also apply to the lost benefits for visitors to the forest. Previous studies proved that forest areas have a positive effect on obtaining psychological relaxation in the people visiting them. However, it was not known whether this restorative experience could be disturbed in any way by the presence of an open dump in the forest. To check how the presence of a landfill affects the visitors, an experiment was planned in which the respondents observed a forest area with a landfill and a forest landscape without a landfill for 15 minutes (control). The respondents then assessed the landscape using the semantic differential method and the Perceived Restorativeness Scale (PRS). An analysis of these observations showed that the presence of a landfill in the forest significantly changed the appreciation of the landscape by the respondents, the values of positive experiences decreased, and the negative experiences increased. Restorativeness was also reduced. Based on the results, it can be concluded that the presence of garbage in the forest may interrupt the restorative experience of its visitors.

## Introduction

There are many studies confirming the positive impact of nature on people and their well-being [1–4]. Visits to nature contribute to the restoration and preservation of mental health [5–7], as well as stress reduction [2,8,9]. Son and Ha [10] proved that increasing contacts with nature serves to improve social and emotional interactions in modern society. Hartig et al. [2], Korpela et al. [11], Tyrväinen et al. [7], Pasanen et al. [1] and Jung et al. [12] showed that people's moods and positive feelings are increasing in natural areas.

With the growing amount of scientific evidence on the therapeutic effects of the forest environment, the importance of forests in promoting the public health of urban dwellers [13], who are more exposed to the loss of contact with nature, is also growing. Limited contact with

**Funding:** The results presented in this paper were obtained as part of a comprehensive study financed by the University of Warmia and Mazury in Olsztyn, Faculty of Agriculture and Forestry, Department of grant No. 30.610.018-110.

**Competing interests:** The authors have declared that no competing interests exist.

nature is referred to as the nature deficit syndrome, a term that has gained widespread use in the literature. It is not, however, recognized by any medical coding schemes such as ICD-10 or DSM-5, the classification and diagnostic tools of the American Psychiatric Association [14], although the resulting disorder has been recognized by Knapp [15] as a disturbing, non-infectious disease of contemporary urban societies. Proximity and accessibility mean that urban and suburban forests are particularly intensively used by society and their social importance continues to grow [16,17]. Research shows that these forests are important to people all year round, often every day [18], they are both a place for individual recreation and recreational and sports activities related to sports competition [18–20].

Some researchers [21–23] believe that human well-being is related to the aesthetics of space, they emphasize the importance of the scenic qualities of a landscape to increase the preferences of visitors. There are many studies relating to the aesthetic qualities of the environment. Research on forest landscape preferences was carried out by, among others, Jensen [24], Vander Stoep and Duniavy [25], O'Leary et al. [26], Janeczko [27], Silvennoinen et al. 2002 [28], Gundersen and Frivold [29], Giergiczny et al. [30] and Filyushkina et al. [31]. The beauty of the landscape, aesthetic values, and the search for aesthetic experiences are important motives for taking up recreational activity in forests [27,32]. However, as noted by Martens et al. [33] and Simkin et al. [34] a preference for a particular environment does not directly mean that it is highly restorative. According to Kaplan and Kaplan's attention restoration (ART) theory [35], only environments that can give a sense of being away (psychological distance from everyday stressors), fascination (the ability to capture involuntary attention), coherence (environmental richness), and compatibility (the ability to fully fulfill the person's intention) will facilitate positive experiences. Thus, not every type of natural environment has a positive effect on restorativeness. Milligan and Bingley [36] believe that the concept of "the natural environment is therapeutic" cannot be accepted without some criticism. Wyles et al. [37] points out that we have considerable knowledge of the benefits of a clean environment, but we know less about how the presence of litter, either in relation to blue (aquatic) or green (terrestrial) environments, can interfere with these benefits [27,38].

Garbage is widely recognized as an ongoing global problem with a number of implications for human health and environmental integrity [39]. Garbage specifically addresses human behavior related to inappropriate waste disposal [40]. In some parts of the world, including Poland, illegal landfills in forests are a big problem [41]. The problem is increasing near urban areas as well as those areas that are generally considered attractive for second home locations. Not only can these illegal dumps pose a threat to wildlife, they can also be dangerous to humans, and their disposal is a heavy financial burden for forest managers. Janeczko [27] and Janeczko and Woźnicka [18] showed that littering is one of the most important factors causing discomfort among tourists and visitors to forests, and thus lowering the assessment of landscape attractiveness. Therefore, garbage is a key factor that may weaken the positive, restorative effects of a clean, natural environment, which is the impact of the forest on humans.

Thereafter, the aim of the study was to answer three scientific questions:

1. To what degree does viewing an open dump located in a forest affect the landscape appreciation (measured by semantic differential technique and perceived restorativeness scale) of healthy volunteers in comparison to conventional control forest environment?

2. Which subscales of perceived restorativeness scale and which adjectives in the semantic differential method indicate the most intense effect of an open dump on volunteers?

3. What do these observations tell us about the nature of the influence of an open dump in the forest on landscape appreciation?

## Materials and methods

### Participants

A group of twenty-four healthy subjects volunteered (mean age 21.63 years ± 1.18 SD, 18 males and 6 females) for research aiming to establish the influence of illegal open dump in the forest on forest landscape appreciation. Volunteers were recruited from the University of Warmia and Mazury in Olsztyn from different fields of study, the recruitment criteria were: 1) to be more than eighteen years old, 2) not have experience with living or working in a forest environment, 3) be healthy. Only participants with Polish nationality were chosen. The study was open-ended; therefore, participants were not discriminated/selected by gender or age.

The participants were divided into two groups, group A and group B and participated in the experiment in these two groups.

This study was ethically reviewed and approved by the Ethical Review Board of the University of Warmia and Mazury in Olsztyn. The number of this ethics statement is 07/2019. All procedures performed in this study were conducted in accordance with the ethical standards of the Polish Committee of Ethics in Science and with the 1964 Helsinki Declaration and its later amendments.

### Experimental stimulation

To test the effect of an open dump in the forest on healthy volunteers, an illegal open dump in the forest was used for the experimental stimulation. For this purpose, an illegal open forest dump was localized in an urban forest in Olsztyn (near the Kortowo Campus: 53˚45'06.2"N 20˚25'10.4"E) (Fig 1). The forest stand with the open dump had a species composition as follows: white oak 80%, white lime 10% and black alder 10% with a medium age of the stand of approximately 90 years. The control forest was located 50m from an illegal open dump in a forest with approximately the same stand age, density and species composition. The open dump had a diameter of 6.5m and contained different kinds of waste, usually cans, bottles,

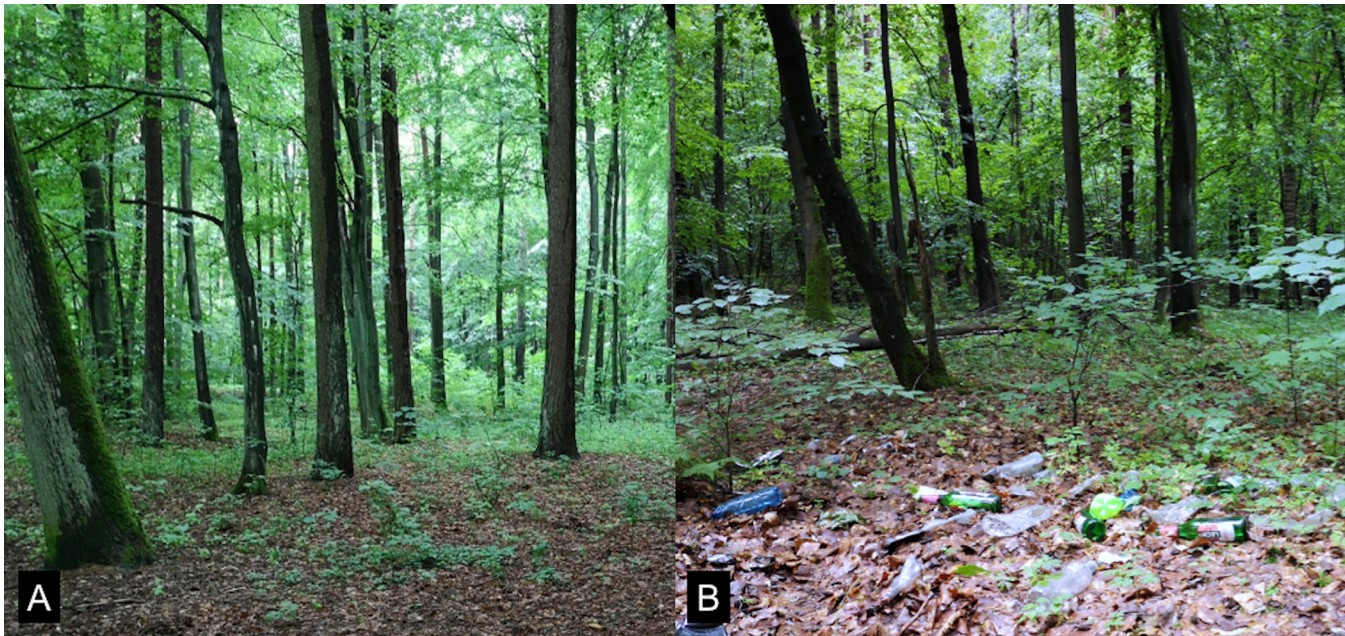

**Fig 1.** View of control forest (A) and forest with illegal open dump (B). Photos were also reported in Bielinis et al. 2020 [78] (not published).

papers and boxes generated after using commercially available products. Because it was the illegal open dump in the forest, the history and origin of the dump was not known to researchers.

The physical environment during these stimulations was recorded (ten recordings of physical environment per each feature).

During visual stimulation of each landscape, each participant looked at the control forest landscape or open dump for 15 minutes, from a distance of 3 meters from the edge of the open dump to eliminate intense olfactory stimulation.

## Measurements

In many experiments, the momentary reaction of a participant was measured before and after stimulation, however, in this study, only landscape appreciation was estimated, thus it was possible to evaluate appreciation by participants only after stimulation by landscape (only post measures were possible). For landscape appreciation, two psychometric tools were applied: the semantic differential method and Perceived Restorativeness scale (PRS).

The Semantic Differential Method [42] is a method previously used in the evaluation of forest landscape appreciation [43,44]. In this method, a list of opposite adjectives, which can describe the quality of environment (impressions which people have in different environments), is evaluated by participants using a Likert scale range from 1 to 7. In this study, by consideration of cultural differences and the applicability for viewing a forest environment and an open dump, a list of 20 pairs of adjectives or adjective-verb pairs were selected.

The Perceived Restorativeness Scale (PRS) [45] is a scale evaluating the restorative properties of the environment. The Polish version contains an adaptation of a list of sixteen items [46,47] evaluated on a 7-point Likert scale. The PRS is divided into four subscales: 'being away', 'compatibility', 'fascination' and 'coherence' (in this version).

## Procedure

The two groups of participants, with the same gender participation in both groups, were involved in the experimental stimulation. For these purposes, on the same day between 10 a.m. and 3:30 p.m. (14th of June 2019) participants in group A and B viewed control forest landscape or forest landscape with an illegal open dump, each group viewed the landscape in reverse order, to avoid the effect of presentation order in the experiment. Before each viewing, participants covered their eyes with a black band for 15 minutes to prevent the stimulation effect by the surroundings (which simulated the effect resetting tents in forest environment [43]). After this resetting, participants viewed the forest landscape (group A) or illegal dumps landscape (group B) for 15 minutes. After this, participants filled in questionnaires o the SDM and PRS scales. The participants then walked to other places, forest (group B) or an illegal dump (group A). Before the second viewing, the participants again covered their eyes to reset the effects of the surrounding landscape for 15 minutes. The next stage was to view the landscape (forest or dump) again for 15 minutes and questionnaires with SDM and PRS scale were then filled in. After the experiment, the participants were informed about the purpose of the experiment. While wearing a black band, viewing the forest and the open dump, the participants were allowed to stand or sit. During the experiment, the participants also filled in other psychometric questionnaires, but those results were not published in this study.

## Statistical analysis and calculation and statistical power

The raw data obtained from the questionnaires were digitalized and analyzed using SPSS 25 Software for Mac (IBM, Armonk, NY, USA). The paired t-test was applied to compare control

forest and open dump landscapes evaluated by each pair of adjectives on the SDM scale and each subscale (or total values) on the PRS scale. The p value obtained in each analysis was corrected with Bonferroni corrections since many features were measured.

The analysis of statistical power was conducted using the free software: G*Power 3.1.9.6 for Mac (Heinrich Hein University, Düsseldorf, Germany). The actual power ($1-\beta$ error probability) was calculated as 0.768. The t-test; means, differences between two dependent means (matched pairs) statistical test was used and a power analysis type of 'Post hoc: Compute achieved power' was applied with an effect size of 0.5 and an $\alpha$ probability error of 0.05.

## Results

### Physical environment

During the experiment, four features of physical environment (Table 1) were measured in each experimental setting (control or waste). Only relative humidity was different in the control than in the waste environment (p = 0.005**) and the other three features were not significant (p from 0.196 to 0.470).

### Landscape appreciation

**Semantic differential method.** The results of landscape evaluation by the semantic differential method of control and waste settings and the results of t-tests are reported in Table 2. Sixteen of pairs of adjectives were significantly different in the waste environment than in the control. The highest significant difference was between the 'dirty-clean' pair: the values were 78.11% lower in the waste setting than in the control (the waste settings were dirtier) and the waste setting was also perceived as 88.51% more chaotic than the control. The environmental differences were not significant, two settings were the same environmental quality in terms of 'dry-wet', 'bright-dark', 'gentle lighting—too bright' and 'quiet-noisy'. As Table 2. shows, the waste setting was significantly more dirty, uncomfortable, unpleasant, ugly, artificial, not enjoyable, insecure, smelly, non-descript, restless, disenchanting, awaking, flat, noisy, closed, and chaotic.

**Perceived restorativeness.** The Perceived Restorativeness Scale results (Table 3) had significantly lower values of total PRS in waste settings than in the control (p < 0.001***). For the subscales, the highest change was in the 'being away' subscale, while on the waste setting this subscale was 62.84% lower than in the control. The second highest change was for 'compatibility', followed by 'fascination' and the lowest change was in 'coherence' (41.99% change). All subscales differed significantly from the control.

**Table 1. Results of the t-test comparison between the control and experimental settings (physical environment); also reported in Bielinis et al. 2020 (not published).**

| Feature | Control | | Waste | | % Δ | t | r | p+ | |
|---------|---------|-----|-------|-----|-----|---|---|----|----|
| | Mean | SD | Mean | SD | | | | | |
| Illuminance (lux) | 3367 | 296.4 | 3074 | 623.7 | 8.72 | 1.344 | 0.040 | 0.196 | |
| Temperature (˚C) | 20.71 | 1.36 | 21.22 | 1.11 | -2.46 | -0.74 | 0.907 | 0.470 | |
| Sound pressure (dB) | 35.84 | 3.09 | 37.05 | 1.682 | -3.38 | -1.09 | 0.671 | 0.292 | |
| Relative humidity (%) | 73.6 | 2.22 | 77.5 | 3.14 | -5.3 | -3.21 | 0.957 | 0.005 | ** |

+ Bonferroni corrected; n = 10

** p < 0.01.

**Table 2. Results of a t-test comparison between the control and waste settings (semantic differential method).**

| Feature | Control | | Waste | | % Δ | t | r | p + | |
|---|---|---|---|---|---|---|---|---|---|
| | Mean | SD | Mean | SD | | | | | |
| Dirty (1)—Clean (7) | 4.75 | 1.29 | 1.04 | 0.99 | 78.11 | -11.16 | 0.008 | <0.001 | *** |
| Uncomfortable (1)—Comfortable (7) | 4.17 | 1.20 | 1.37 | 1.13 | 67.15 | -7.43 | -.239 | <0.001 | *** |
| Unpleasant (1)—Pleasant (7) | 5.88 | 1.11 | 2.38 | 0.87 | 59.52 | 13.34 | .184 | <0.001 | *** |
| Ugly (1)—Beautiful (7) | 5.79 | 0.72 | 2.50 | 0.72 | 56.82 | 13.90 | -.292 | <0.001 | *** |
| Artificial (1)—Natural (7) | 6.54 | 0.50 | 2.83 | 1.09 | 56.73 | 15.66 | .091 | <0.001 | *** |
| Not enjoyable (1)—Enjoyable (7) | 5.42 | 1.17 | 2.38 | 1.01 | 56.09 | 9.01 | -.137 | <0.001 | *** |
| Insecure (1)—Secure (7) | 4.96 | 1.23 | 2.38 | 1.01 | 52.02 | 6.40 | -.544 | <0.001 | *** |
| Smelly (1)—Odorless (2) | 5.54 | 0.93 | 3.04 | 0.95 | 45.13 | 7.85 | -.369 | <0.001 | *** |
| Non-descript (1)—Unique (7) | 5.29 | 1.12 | 2.92 | 1.21 | 44.80 | 5.65 | -.556 | <0.001 | *** |
| Restless (1)—Calm (7) | 4.92 | 1.13 | 2.75 | 1.07 | 44.11 | 6.30 | -.160 | <0.001 | *** |
| Disenchanting (1)–Enchanting (7) | 5.46 | 1.17 | 3.08 | 0.97 | 43.59 | 6.25 | -.489 | <0.001 | *** |
| Awaking (1)—Soothing (7) | 4.00 | 1.41 | 2.79 | 0.77 | 30.25 | 3.94 | .158 | 0.017 | * |
| Flat (1)—Three dimensional (7) | 5.79 | 0.72 | 4.21 | 1.31 | 27.29 | 6.05 | .322 | <0.001 | *** |
| Dry (1)—Wet (7) | 3.67 | 1.43 | 3.17 | 1.27 | 13.62 | 2.63 | .769 | 0.392 | - |
| Bright (1)—Dark (7) | 4.00 | 0.88 | 3.46 | 0.88 | 13.50 | 3.41 | .612 | 0.063 | - |
| Gentle lighting (1)—Too bright (7) | 2.79 | 0.93 | 3.17 | 0.91 | -13.62 | -1.74 | .348 | 1.000 | - |
| Quiet (1)—Noisy (7) | 2.71 | 1.12 | 3.46 | 1.06 | -27.68 | -2.58 | .153 | 0.431 | - |
| Pleasing sound (1)—Irritating noise (7) | 2.50 | 1.10 | 3.75 | 1.18 | -50.00 | -5.71 | .564 | <0.001 | *** |
| Open (1)—Closed (7) | 3.08 | 1.06 | 4.67 | 1.23 | -51.62 | -7.32 | .585 | <0.001 | *** |
| Orderly (1)—Chaotic (7) | 2.96 | 0.99 | 5.58 | 1.21 | -88.51 | -6.74 | -.481 | <0.001 | *** |

+—Bonferroni corrected; Likert scale of seven stages was used for this questionnaire.

\*\*\*p< 0.001

\*\*p < 0.01

\*p < 0.05 paired t-test; n = 24.

## General discussion

The experiment took into account two forest fragments that differed to a small extent due to the analyzed bioclimatic parameters. Illumination of the area was a key aspect that could influence the obtained results. Many studies show that light and its access to the forest floor can affect how the forest is perceived. For example, Takayama et al. [43], comparing the results of

**Table 3. Results of t-test comparison between control and waste settings (Perceived Restorativeness Scale).**

| Feature | Control | | Waste | | | t | r | p + | |
|---|---|---|---|---|---|---|---|---|---|
| | Mean | SD | Mean | SD | % Δ | | | | |
| Being away | 5.14 | 1.02 | 1.91 | 0.95 | 62.84 | 11.25 | -0.01 | <0.001 | *** |
| Compatibility | 4.8 | 1.16 | 2.04 | 0.07 | 57.50 | 9.869 | 0.004 | <0.001 | *** |
| Fascination | 5.31 | 0.98 | 2.75 | 0.67 | 48.21 | 11.15 | 0.111 | <0.001 | *** |
| Coherence | 5.43 | 0.63 | 3.15 | 0.98 | 41.99 | 9.925 | 0.078 | <0.001 | *** |
| PRS Total | 5.19 | 0.81 | 2.57 | 0.57 | 50.48 | 12.64 | -0.06 | <0.001 | *** |

+—Bonferroni corrected

\*\*\*p< 0.001

\*\*p < 0.01.

\*p < 0.05 paired t-test; n = 24.

the SD scale for respondents' impressions about different forest environments, confirmed more positive assessments for thinning forests. Takayama [48] showed that in unmanaged forests, thinning can improve forest environments, which can improve users' impressions and evaluations of forests. Traits of restorative environments could be improved by conducting forest management. Sparse forests are preferred due to the greater transparency of the forest floor, they are perceived as more visually attractive [24,26,49,50]. In the current research, however, the dump was not an open space, it was located under a canopy of trees, inside the stand. Probably for this reason, in the case of "bright—dark" or "gentle lighting—too bright" no differences were confirmed by the presence or absence of a landfill. There were also no differences observed for the terms "quiet—noisy" and "dry—wet". The chosen environments did not differ practically in the level of sounds and the statistical difference noted in the case of air humidity was not large enough to differentiate the respondents' assessments. Overall, the respondents rated the forest without a landfill more positively. The two environments differed the most in the terms "clean-dirty" and "tidy-chaotic". This is an interesting observation because it proves that one values a space that is perceived as orderly. Perhaps, therefore, not only thinning out, access to light as well as the ordering of space is of key importance for the perceived restorativeness of forests. Sugiyama [51] reached similar conclusions, stating that naturalness does not guarantee high preference. What people prefer is likely to be tidy (and natural) scenes. The relation between naturalness and preference seems complex: a positive influence from natural elements and a negative influence from untidy appearance. Thompson et al. [52] found that people would like a compromise between a very wild woodland and a woodland park: partly managed areas but also a very natural, feel, without obvious evidence of human influence where possible. Other studies [33,53] point out that the "orderly" forest is more preferable and has better regenerative properties than a forest that is perceived as wild and all-natural. This is important information for forest managers and local administrators, especially if significant factors influencing the use of forest areas include, next to the forest, the lack of litter in the forest [52]. Littering of the site is a reason to limit visits to a given site and limit the use of recreational infrastructure [54–56].

This study is part of a research project on factors depreciating the forest landscape. So far, much attention has been paid to phenomena responsible for chaos, lack of order in the forest space (such as traces of forest management, e.g. stumps, logging waste, forest crops [57,58], deadwood (e.g. [29,59–61]) or piston [62–64]). As the current study shows, garbage is also an element that depreciates the forest landscape. The forest with a wild dump was perceived as much dirtier and more chaotic. The current research showed that terms such as "comfortable—uncomfortable", "friendly—unfriendly", "ugly—beautiful", "artificial-natural", "non-enjoyable—enjoyable" were also significantly different. The participants in the experiment used negative terms to describe the dump (unpleasant, uninvolved, ugly, artificial, restless, etc.). The presence of rubbish influenced both the development of negative emotions in the respondents (e.g. more insecure, more uncomfortable) and the negative perception of landscape features by the respondents (e.g. flatter, uglier, dirtier), while the features of the physical environment mostly did not change (did not differ significantly). This means that the forest environment with an existing garbage dump is naturally perceived by the respondents as an environment with negative features both for the emergence of emotions and from the point of view of landscape properties. However, based on the conducted research, it is difficult to determine why this is so, probably due to cultural and biological conditions. The current study shows a clear relationship between many difficult-to-perceive, subjective aesthetic values of space and spatial order. In architecture and urban planning, landscape shaping is directly connected with the problems of spatial order and indirectly with spatial planning [65]. This claim also applies to the forest landscape. The aesthetics of the forest landscape and the way people perceive the

forest landscape is conditioned by a sense of internal cohesion, spatial order and lack of elements that depreciate the landscape. This was confirmed by the current results on the perception of restorativeness (PRS).

The results obtained from the PRS analysis also indicate that the perception of a forest without a landfill differed significantly from that with garbage. In all four subscales, statistically significant differences were noted, the largest with regard to "being away". Other studies [46,66,67] indicated that "coherence" was rated higher than the other three components of perceived restorativeness. Tennegart Ivarsson and Hagerhall [67] analyzed the restorativeness and preferences of two types of gardens (more organized and rural with free plant composition). Korpela and Hartig [66] compared favorite and unpleasant places identified by Finnish students and Hauru et al. [46] open, semi-closed and close view to the housing matrix, to the road matrix and the urban matrix. Therefore, these were not studies comparing such radically different environments as in the current study. Coherence in the current research turned out to be the most important feature of restorativeness for both environments. However, in general terms, it was found that this subscale is less important than "being away", "compatibility" or "fascination." The experience of coherence in the case of a landfill was low and was significantly different (41.99%) from a pure forest in which no garbage may have been perceived as distracting the recipient from the scope and richness that the environment has to offer. It may also indicate that there was an understanding among respondents that the studied fragment of the forest with garbage was a specific episode in time and space and it does not belong to a larger whole and is not a representative feature for the urban forests around Olsztyn. Garbage is a factor that distorts the expected remedial effects of the forest environment. According to Korpel and Hartig [66] restoration can proceed when four factors characterize the person-environment interaction, apart from coherence, they also include, for example, being away. Being away involves a psychological and possibly geographical distance from one's usual context, including the work one ordinarily does and the pursuit of particular goals and purposes. The feeling of "being away" means, as intended by Kaplan and Kaplan [68], a psychological distance from everyday stressors. This distance increases with immersion into recreation. In turn, the quality and comfort of rest, and therefore detachment from stress, depends on the aesthetics of the space. The sensual experience of nature is an important theme for outdoor recreation. The results of Zeidenitz et al. [69] show that people treasure the experience of nature and beautiful landscapes in outdoor activities. This is the main theme for all outdoor recreational activities, as is "relaxation and wellness". A study by O'Brien and Forster [70] shows that beauty, scenery, wildlife, sensory and seasonal experiences, a sense of freedom, detachment from everyday life and the atmosphere of woodlands were key factors for participants in the experience of being active in the open. According to the current study, rubbish in the forest significantly reduces the aesthetics of the forest and thus reduces the recreational value of the forest environment. Hence, the largest difference in the PRS subscales was noted in the case of "being away".

Relaxation and restorativeness are supported by "fascination", another subscale of the PRS. Nature is certainly well-endowed with fascinating objects, as well as offering many processes that people find engrossing [35]. Kaplan and Kaplan define fascination as [68] the ability for effortless attention. According to Kaplan [35], fascination is a central component of a restorative experience. That is not to say, however, that the presence of fascination guarantees that directed attention can rest. Fascination is a necessary, but not sufficient basis for recovering directed attention. The current study shows that fascination was, after consistency, the most important subscale of restorativeness of both environments, both the forest and the landfill and the forest constituting the control sample. As predicted, fascination with the environment with the landfill was much lower than the fascination with the forest without rubbish. The

sight of the rubbish attracted the attention of the participants, although it was not as positive as the natural conditions as the focus was on the rubbish, usually distracting the viewer from the beautiful scenery around them. Overall, this subscale was less important than "being away" or "compatibility." Hartig et al. [2] points out that fascination, renewal and other positive motivations can be used to promote ecological behavior. Paying attention to the direct and indirect effects of fascination, but also being away, coherent and consistent, can help to prioritize actions that promote environmental behavior. For example, people who engage in valuable outdoor activities, rather than consuming or motorized recreation are more likely to be pro-environmental [71]. Their outdoor recreation is probably more closely related to the interests of the natural environment [2].

"Compatibility" was a strongly differentiating restorativeness of both environments. The natural environment is experienced as particularly high in compatibility [35]. The forest is perceived by many people, especially city dwellers, as a place where you can relax and rest. If the environment matches the needs, the goals that a person wants to achieve are viewed as highly compatible. A forest without rubbish was experienced as compatibility, but both "coherence", "being away" and "fascination" were more important than this. In the case of a landfill environment, coherence and fascination were of greater importance. The littered space was not in line with the behavioral goals of individuals.

## Future research

So far, many studies have been prepared that conclusively prove the positive impact of nature on human relaxation. However, not much is known about the impact of negative manifestations of human activity in the environment (e.g. landfills, post-mining heaps, ash dumps) on restorative and human well-being. The current work focused on the impact of forest rubbish on the regenerative properties of the forest. The forest landscape is, by nature, favored for recreation, it is perceived as very attractive, and its positive influence on humans has also been proven many times. In the future, it is worth examining to what extent the results of the study also apply to other landscape variants (water, agricultural, arranged greenery, etc.). Thanks to this, it will be possible to understand even better the mechanism of the impact of garbage on humans and within which psychological scales this phenomenon is most visible. It is certainly worth researching how the nature of the garbage and what elements it consists of affects the restorative mechanism. This thread seems to be particularly important in the context of forestry. Forest management generates a lot of natural waste. Intensive harvesting and thinning works in the forests lead to temporary clutter in the forest. A forest with wood waste, wood chips, broken branches, etc. can be perceived by people not only as a chaotic, messy space, but also as a littered space, full of waste, reducing its rebuilding properties. Therefore, it is worth examining the differences in restorativeness between the space with natural elements of litter and the forest with anthropogenic rubbish (paper, plastic, rubber, bulky rubbish, etc.).

Artificial, wild garbage dumps in the forest are created in various places in the forest, usually along forest roads in the vicinity of water reservoirs, but garbage also appears in areas that are intensively used for recreation (e.g. a leisure clearing, a clearing with picnic areas). From the point of view of the impact of garbage on the restorativeness and appreciation of the landscape, the genesis of garbage (illegal landfill / insufficient amount of waste bins) is less important at this point, the context of the place is certainly more important. There is also room for further exploration of this phenomenon. To what extent are the rhetorical properties of a landfill in a forest with a water reservoir lower than that generated by a landfill in a forest adjacent to a picnic clearing? At this point, another question arises regarding the concentration of garbage. Is the volume of rubbish in a given place and the experience of differentiated spaces due

to the concentration of rubbish important for the appreciation of the landscape and does it affect the degree of restorativeness of forest visitors? This thread also requires further investigation, the more so as, for example, in urban planning, there has been a long-standing view that if there are disharmonious elements in the analyzed area and its aesthetic values have already been degraded, further negative actions will not cause a sharp increase in pejorative feelings. Without further research on the phenomenon of waste concentration, we will not be able to say whether this view also applies to the environment. The level to which the environment is favorable and garbage harmful may be the result of an individual's emotional connection with nature. For example, Mayer et al. [72] found that people who were more connected with nature, i.e. had a stronger emotional bond with the natural world, received greater benefits from visits to nature than those who were less related. Does it also mean that people with a strong relationship with nature experience the effects of littering more deeply, that garbage is a greater stress factor for them than for those who have a less connection with nature? Probably yes, but this hypothesis must be proven.

In addition, the issue of the impact on garbage and people's attachment to a given place also needs to be resolved. The existing arrangements in this regard are quite general and divergent. For example, Kyle et al. [73] proved that people with greater attachment to a given place are more sensitive to rubbish than people with a weaker connection with nature. The work of other researchers (e.g. [74]) has shown that people with a stronger bond have a greater adaptability and are therefore able to overlook this negative impact. In future research, therefore, it is worth paying more attention to socio-demographic aspects, the more so as there are also studies showing that elderly people are more sensitive to problems related to garbage and maintenance, and therefore their reactions to garbage may differ from those established for the younger part of the population. In future research, it is also worth examining in more depth the relationship between the intensity and frequency of visits by experiment participants to the forest and the benefits they obtain from experiencing the forest environment. Subiza-Pérez et al. [75] believe that more visits can lead to habituation to the environment, which therefore generates less interest, fascination or aesthetic pleasure. Consequently, reduced immersion and engagement would hinder—at least to some extent—the speed of playback and refresh (more visits could lead to habituation to the environment that would therefore trigger less interest, fascination, or aesthetic enjoyment). Consequently, the reduced immersion and engagement would hinder—at least to some extent—the rates of recovery and refreshment. Paying attention to the direct and indirect effects of fascination, being away, and consistency and compliance can help you prioritize actions that promote ecological behavior. In particular, these findings encourage a focus on fascination because people who spontaneously immerse themselves in environmental experiences are more prone to pro-environmental behavior [71].

## Limitations

The biggest limitation of the experiment was the relatively small sample of respondents, 24 people were involved in the study, which allowed conducting an experiment with a power equal to 0.768, which is on the border of an acceptable value in social sciences. In other studies of this type, much less numerous groups of respondents were involved, and significant conclusions were drawn [48,76,77]. Another limitation may be the lack of an additional control group in which the research would look at the anthropogenic landscape, e.g. urban landscape, it would allow a better understanding of the processes taking place during landscape assessment. However, the current system, with one control in the form of a forest, is also acceptable, in the authors' opinion, at the current stage of research, which includes research on the impact

of rubbish on the appreciation of the landscape. The limitation of the research may also be the evaluation of the influence of the presented landscapes on emotions, but it was examined in this context in another work that was submitted for publication [78].

## Conclusions

Forests provide society with many health benefits. A growing body of evidence points to a positive relationship between forest exposure and the mental well-being and physical health of people, especially urban residents. It turns out, however, that not every forest is fully restorative. Garbage is one of the factors that depreciate the forest landscape, and at the same time, a factor disturbing the expected remedial effects of the forest environment. The current study shows evidence that the presence of garbage can undermine the psychological benefits that clear forest views usually provide. The forest rubbish was perceived as an environment of lower restoration quality than natural forest scenery. This is evidenced by the results of the DS method used, as well as the PRS subscales, such as coherence, being away, fascination and compatibility. This is important information not only from a cognitive but also a practical point of view. Establishing the relationship between the forest experience and the psychological benefits of staying in the forest is important both in terms of design and management of various types of outdoor environments, but also in terms of shaping pro-ecological behavior, leading to greater responsibility for the environment. The feeling of confusion, lack of curiosity, nervousness or lack of belonging to a given place caused by exposure to a littered forest significantly reduces the restorative benefits associated with a visit to the forest but also, in a broader perspective, limits the ability to understand and appreciate the importance of the forest for human well-being.

## Author Contributions

**Conceptualization:** Ernest Bielinis, Emilia Janeczko, Natalia Korcz, Lidia Bielinis.

**Data curation:** Ernest Bielinis, Emilia Janeczko, Krzysztof Janeczko, Lidia Bielinis.

**Formal analysis:** Ernest Bielinis, Emilia Janeczko, Krzysztof Janeczko, Lidia Bielinis.

**Funding acquisition:** Ernest Bielinis, Emilia Janeczko, Natalia Korcz, Krzysztof Janeczko, Lidia Bielinis.

**Investigation:** Ernest Bielinis, Emilia Janeczko, Krzysztof Janeczko, Lidia Bielinis.

**Methodology:** Ernest Bielinis, Emilia Janeczko, Krzysztof Janeczko, Lidia Bielinis.

**Project administration:** Ernest Bielinis, Emilia Janeczko, Natalia Korcz.

**Resources:** Ernest Bielinis, Emilia Janeczko, Natalia Korcz, Lidia Bielinis.

**Software:** Ernest Bielinis, Emilia Janeczko.

**Supervision:** Ernest Bielinis, Emilia Janeczko.

**Validation:** Ernest Bielinis, Emilia Janeczko, Lidia Bielinis.

**Visualization:** Ernest Bielinis, Emilia Janeczko, Lidia Bielinis.

**Writing – original draft:** Ernest Bielinis, Emilia Janeczko, Lidia Bielinis.

**Writing – review & editing:** Ernest Bielinis, Emilia Janeczko, Natalia Korcz, Krzysztof Janeczko, Lidia Bielinis.

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
