## [Decision Letter · Decision Letter 0]

9 Jun 2022

PONE-D-21-25220

Effect of an Illegal Open Dump in an Urban Forest on Landscape Appreciation: A Pilot Study

PLOS ONE

Dear Dr. Bielinis,

Thank you for submitting your manuscript to PLOS ONE. After careful consideration, we feel that it has merit but does not fully meet PLOS ONE’s publication criteria as it currently stands. Therefore, we invite you to submit a revised version of the manuscript that addresses the points raised during the review process.

Please note that we have only been able to secure a single reviewer to assess your manuscript. We are issuing a decision on your manuscript at this point to prevent further delays in the evaluation of your manuscript. Please be aware that the editor who handles your revised manuscript might find it necessary to invite additional reviewers to assess this work once the revised manuscript is submitted. However, we will aim to proceed on the basis of this single review if possible.

The reviewer has raised a number of concerns in the attached annotated manuscript.

Could you please revise the manuscript to carefully address the concerns raised?

We look forward to receiving your revised manuscript.

Kind regards,

Sebastian Shepherd

Staff Editor

PLOS ONE

Journal Requirements:

2. Thank you for stating the following financial disclosure: "The funders had no role in study design, data collection and analysis, decision to publish, or preparation of the manuscript." 

4. Please ensure that you refer to Figure 1 in your text as, if accepted, production will need this reference to link the reader to the figure.

5. We note that Figure 1 includes an image of a participant in the study. 

7. Please provide field permits and for reproducibility purposes to specify the location of the different fields sites assessed.

Reviewers' comments:

Reviewer's Responses to Questions

**Comments to the Author**

1. Is the manuscript technically sound, and do the data support the conclusions?

Reviewer #1: Yes

2. Has the statistical analysis been performed appropriately and rigorously? 

Reviewer #1: Yes

3. Have the authors made all data underlying the findings in their manuscript fully available?

Reviewer #1: Yes

4. Is the manuscript presented in an intelligible fashion and written in standard English?

Reviewer #1: Yes

5. Review Comments to the Author

Reviewer #1: Please find the attached pdf document.

6. PLOS authors have the option to publish the peer review history of their article (what does this mean?). If published, this will include your full peer review and any attached files.

Reviewer #1: No

---

## [Author Response · Author response to Decision Letter 0]

20 Oct 2022

RED – Our response

BLACK – Editor comments 

Rebuttal letter

In the following article, we change as follows:

1. We suited the authors’ list to the format of PLOS ONE journal (PO).

2. We formatted the style of the manuscript to PO – we changed the size of the fonts in some parts and unbolded them in some regions.

3. We added geolocalisation data for the measured forest for identification of the site.

4. We carefully checked the literature and used the Mendeley desktop to format the PO style.

5. We moved two additional photographs and changed the abbreviation of figure 2.

6. We add the information about data availability – the data are available in the office of the primary author.

7. We changed Figure 1 – the photos with people were removed.

Detailed changes, step by step, followed by Editor comments:

Line 2 – we removed the sentence in the title “A pilot study”

Line 4-20 – we prepared a list of authors following PLOS ONE style

For all manuscript – we change the letters to style ‘Times New Roman

All manuscript – we changed the citation style in Mendeley to the style of PLOS ONE

Line 113 – We change the big letter to small in “Methods’

Line 133 – We added geolocalisation data for the forest site

Line 180 – We change the big letter to small in “Analysis”

Line 200-201 – We change the formatting of subtitles and change the big letter for small in ‘appreciation’ and in ‘difference’ words

215 – We change the formatting of the subchapter name

242 – We change the formatting of the subchapter name

Line 364 - We change the formatting of the subchapter name and change the letter from big to small in ‘Research’

Line 689 – We change the size of the letters describing Figure 1 to big letters, and we divided the sentence into two sentences.

2. Thank you for stating the following financial disclosure: "The funders had no role in study design, data collection and analysis, decision to publish, or preparation of the manuscript." 

There is no funding. We mark this during submission.

No role of funders, we mark this in a form.

No special fundings, we mark this in a form.

We included this hear.

Figure: We actualize the file with a photo of the control and experimental site

Line 464-686 – We automatically change the formatting of literature to PLOS ONE in Mendeley

We added anonymised data here (in Harvard repository):

Bielinis, Ernest, 2022, "Replication Data for: A Dataset of emotions measurements was conducted in the forest interrupted by an open dump (rubbish) and in the control forest.", https://doi.org/10.7910/DVN/53VSL2, Harvard Dataverse, V1

4. Please ensure that you refer to Figure 1 in your text as, if accepted, production will need this reference to link the reader to the figure.

Yes, We refer to figure 1 in the text.

5. We note that Figure 1 includes an image of a participant in the study. 

We removed part of Figure 1 and added a fixed figure.

The list is complete and correct. We change the formats in the Mendeley.

7. Please provide field permits and for reproducibility purposes to specify the location of the different fields sites assessed.

We added geolocalsiation data of forest in line 133.

…

Thank you!

---

## [Decision Letter · Decision Letter 1]

4 Nov 2022

Effect of an Illegal Open Dump in an Urban Forest on Landscape Appreciation

PONE-D-21-25220R1

Dear Dr. Bielinis,

We’re pleased to inform you that your manuscript has been judged scientifically suitable for publication and will be formally accepted for publication once it meets all outstanding technical requirements.

Kind regards,

Peter Edwards

Academic Editor

PLOS ONE

Additional Editor Comments (optional):

Reviewers' comments:

Reviewer's Responses to Questions

**Comments to the Author**

1. If the authors have adequately addressed your comments raised in a previous round of review and you feel that this manuscript is now acceptable for publication, you may indicate that here to bypass the “Comments to the Author” section, enter your conflict of interest statement in the “Confidential to Editor” section, and submit your "Accept" recommendation.

Reviewer #2: All comments have been addressed

2. Is the manuscript technically sound, and do the data support the conclusions?

Reviewer #2: Yes

3. Has the statistical analysis been performed appropriately and rigorously? 

Reviewer #2: Yes

4. Have the authors made all data underlying the findings in their manuscript fully available?

Reviewer #2: Yes

5. Is the manuscript presented in an intelligible fashion and written in standard English?

Reviewer #2: Yes

6. Review Comments to the Author

Reviewer #2: Upon reading the revised manuscript, I see that the authors have addressed all the previous reviewer comments. This work is original, addresses previous research, the methods are reproducible with statistical analysis and control, and it makes a contribution to related current and future research.

7. PLOS authors have the option to publish the peer review history of their article (what does this mean?). If published, this will include your full peer review and any attached files.

Reviewer #2: No

---

## [Editor Report · Acceptance letter]

11 Nov 2022

PONE-D-21-25220R1 

Effect of an Illegal Open Dump in an Urban Forest on Landscape Appreciation 

Dear Dr. Bielinis:

I'm pleased to inform you that your manuscript has been deemed suitable for publication in PLOS ONE. Congratulations! Your manuscript is now with our production department. 

Kind regards, 

on behalf of

Dr. Peter Edwards 

Academic Editor

PLOS ONE